# Research on High Robustness Underwater Target Estimation Method Based on Variational Sparse Bayesian Inference

**Libin Du, Huming Li, Lei Wang** , **Xu Lin and Zhichao Lv** *

College of Ocean Science and Engineering, Shandong University of Science and Technology, Qingdao 266590, China; dulibin@sdust.edu.cn (L.D.); 2017051211@hrbeu.edu.cn (H.L.); skd996159@sdust.edu.cn (L.W.); linxu@sdust.edu.cn (X.L.)
* Correspondence: lvzhichao@hrbeu.edu.cn

**Abstract:** Pulse noise (such as glacier fracturing and offshore pile driving), commonly seen in the marine environment, seriously affects the performance of Direction-of-Arrival (DOA) estimation methods in sonar systems. To address this issue, this paper proposes a high robustness underwater target estimation method based on variational sparse Bayesian inference by studying and analyzing the sparse prior assumption characteristics of signals. This method models pulse noise to build an observation signal, completes the derivation of the conditional distribution of the observed variables and the prior distribution of the sparse signals, and combines Variational Bayes (VB) theory to approximate the posterior distribution, thereby obtaining the recovered signal of the sparse signals and reducing the impact of pulse noise on the estimation system. Our simulation results showed that the proposed method achieved higher estimation accuracy than traditional methods in both single and multiple snapshot scenarios and has practical potential.

**Keywords:** array signal processing; sparse Bayesian learning; direction estimation; pulse noise

## 1. Introduction

As an underwater sensor, a hydrophone can realize the real-time monitoring of various opportunistic sound sources and environmental noise in the ocean. However, estimating the distance and direction of the sound source solely based on a single hydrophone is difficult, and a single hydrophone has a low signal-to-noise ratio (SNR) and limited detection range. On the contrary, an array of hydrophones exhibits strong capabilities in estimating the direction and distance of the sound source and has a significantly higher SNR than a single hydrophone. Research on underwater acoustic arrays in terms of SNR improvement, distance, and direction estimation has become a significant topic.

Array signal processing is a technology that uses a group of sensor arrays to spatially sample signals and then uses corresponding signal processing algorithms to enhance and estimate the parameters of the received data. Compared to methods that use a single sensor to collect and process signals, array signal processing technology can achieve spatial gains by utilizing the spatial characteristics of the signal, thereby improving the accuracy of parameter estimation [1].

DOA estimation of underwater wave propagation is a significant research subject in array signal processing and has substantial theoretical and practical implications for underwater target detection and tracking. In the field of underwater acoustic signal processing, commonly used DOA estimation methods include conventional beamforming and the Multiple Signal Classification (MUSIC) algorithm. However, these methods have limited ability to estimate the direction of adjacent signal sources and may even fail in the case of impulsive noise [2]. Therefore, conducting research on the DOA estimation of underwater targets has significant theoretical and practical implications.

Beamforming technology, based on array technology, is the main approach for high-precision target detection. In their research, Kawachi et al. demonstrated the design and testing of an echo-PIV system that efficiently mapped the interior and fluid flow of a submerged vessel using a single divergent signal wave and delay-and-sum processing. However, the DW-DAS echo-PIV method is not useful for sensing leakage points and underwater debris at relatively short distances and over a narrow field of view [3]. Meanwhile, Shostak et al. proposed a new method for estimating the distance to any underwater object or physical phenomenon by analyzing the curvature of the wavefront and its impact on the measuring sonar system of correlated noise. They also provided results that substantiated this method [4]. Li et al. demonstrated a method for estimating seabed parameters that used the spatial characteristics of the ocean's ambient noise without relying on matched-field processing [5]. Zhou et al. reported improvements over conventional PGC methods, and the hydroacoustic sensor system has great potential in large-scale multiplexing [6]. Li et al. investigated the effectiveness of array processing for the passive monitoring of gas seeps and proposed using beamforming methods to enhance the SNR and improve the productivity of passive acoustic systems [7]. Verdon et al. presented a case study showcasing the use of an "L"-shaped downhole fiber-optic array for monitoring microseismic activity [8]. Other influential work includes Schinault et al., 2019. The array, in its current state of development, is a low-cost alternative to obtain quality acoustic data from a towed array system. Their study demonstrated that this array could be used for observing whales and ship tonals at ranges up to 5 km, receiving acoustic signals from targets of interest with enhanced SNR and directional sensing capabilities. Marine mammal vocalizations have been captured by this prototype array, and whale species have been identified through visual observation [9]. Xie et al. proposed a robust wideband beamforming algorithm based on subspace spectrum separation for addressing the issue of manifold deviations that may occur in sensor arrays. In this algorithm, a sensor array manifold calibration method based on subspace partitioning was proposed, and an improved interference–noise covariance matrix reconstruction method based on spectral separation was derived. Firstly, the space was divided into several subspaces using the Capon spatial spectrum, and the array manifold deviation was calibrated. Then, noise and interference information was accurately extracted through spectral separation, and finally, the optimal beamformer was designed based on the extracted information. Simulation results showed that this algorithm had good performance for different types and ranges of array manifold deviations. Furthermore, the wideband interference will be further considered in future work [10]. Zou et al. developed a hybrid analytical–numerical method that combines the analytical technique with the acoustic superposition approach to predict the sound radiation of a spherical double-shell within the ocean's acoustic environment. Green's function was utilized to simultaneously analyze the coupled vibration of fluid–structure, near-field, and far-field sound radiation. To reduce the computational complexity, the near-field was simulated using the image source method, while the far-field was simulated using the normal mode method. This method was used to calculate the sound radiation field of a spherical double-shell with positive and negative gradient sound velocity profiles in a shallow ocean acoustic environment. However, there was no obvious interfering phenomenon in the contour of the sound pressure distribution when the spherical shell was at a certain submerging depth. This requires further study of the related mechanism [11]. The numerical results were compared with finite element calculation results, and the efficiency was improved without compromising the calculation accuracy. A deconvolution method for conventional beamforming (CBF) was proposed in reference [12], which showed theoretically higher array gain (AG) than CBF and provided the possibility of detecting weak signals using the SNR [12]. However, simulation data processing showed that effective AG decreased with a decreasing SNR. The method of output signal subspace deconvolution for CBF was used to recover most of the AG loss and track the azimuth and time of weak signals. Frequency difference beamforming (FDB) provided a robust estimate of the wave propagation direction by shifting the signal processing to lower frequencies.

Xie et al. proposed a deconvolution frequency difference beamforming (Dv-FDB) method to improve array performance, which produced narrower beams and lower sidelobes while maintaining robustness. Based on this, the R-L algorithm was used for deconvolution to make Dv-FDB's spatial spectrum clearer. Simulation and experimental results showed that Dv-FDB was superior to FDB in higher resolution and lower sidelobes while maintaining robustness. Existing R-L methods are limited to arrays with offset-invariant beam patterns [13]. Byun et al. (2020) proposed a multi-constraint method for matching field processing (MFP) to address the uncertainty of the array tilt, and the experimental results verified the robustness of MFP. In summary, beamforming-based target direction estimation algorithms for underwater acoustic arrays can improve the performance of weak signal detection and underwater noise suppression, but the computational complexity of these algorithms needs to be considered. An additional important source of mismatch is the array tilt, which has not received much attention, in spite of its significant impact, especially for a large array tilt observed in shallow environments [14]. Other influential works in this field included Zhu et al. and Zhang et al. [15,16].

Another representative class of estimation algorithms is the MUSIC algorithm. MUSIC is a high-resolution DOA estimation algorithm that was first introduced by Schmidt in 1986. It is a non-parametric algorithm that does not require any prior knowledge of signal statistics, and it is widely used in many fields, including radar, sonar, and wireless communications. The main idea behind the MUSIC algorithm is to transform the received signal into the frequency-domain and estimate the DOAs of the incoming signals based on the eigenvalues and eigenvectors of the covariance matrix of the received signal. Specifically, the MUSIC algorithm first divides the entire space into two subspaces: the signal subspace and the noise subspace. The signal subspace contains the eigenvectors corresponding to the signal, while the noise subspace contains the eigenvectors corresponding to the noise. The DOAs of the incoming signals are then estimated by calculating the peaks of the spectrum of the noise subspace. Compared to other DOA estimation algorithms, such as beamforming and the Estimation Signal Parameter via Rotational Invariance Techniques (ESPRIT), MUSIC has several advantages, including a high-resolution, robustness to noise, and the ability to handle both coherent and incoherent signals. However, it also has some limitations, such as sensitivity to array geometry, the need for an accurate estimation of the noise subspace, and computational complexity. Overall, MUSIC is a powerful and widely used DOA estimation algorithm that has applications in many fields, including signal processing, wireless communications, radar, and sonar.

Yi et al. utilized passive array sonar systems to track a changing number of underwater targets, also known as acoustic emitters [17]. However, the authors did not consider information fusion among multiple passive sonar's systems. Huang et al. addressed the problem of DOA estimation with one-bit quantized array measurements. Otherwise, the approximation error becomes relatively large at a high SNR, which deserves further Investigation [18]. Cheng et al. proposed a marine environment noise suppression method for multiple-input multiple-output (MIMO) applied to the DOA estimation of multiple targets. In future work, it is worth exploring further optimization of the noise suppression algorithm model to reduce the impact of pre-estimation results on the DOA estimation accuracy [19]. As the underwater detection platform has a limited size, the traditional bulky linear array is not feasible. To address this issue, Li et al. investigated the joint processing–MUSIC (JMUSIC) algorithm for estimating the DOA in shallow sea multi-path environments using a non-uniform line array of acoustic vector sensors. It is a pity that the authors only conducted research in an ideal situation and did not take into account complex situations [20]. Zhu et al. proposed a method for obtaining the optimal waveform estimation of source signals in a spatial scanning orientation through the estimation of the maximum posterior probability criterion and the iterative convergence process of the constraint equation. The experiment yielded excellent results in the case of single snapshots, but it is also worth paying attention to how fast the shots were [21]. Ahmed et al. conducted a comparative study of deterministic and heuristic algorithms for viable

DOA estimation for different dynamic objects in underwater environments. To achieve the precise positioning of underwater targets at a close range [22], Ahmed et al. utilized the Cuckoo Search Algorithm (CSA) and swarm intelligence to optimize DOA estimation with a Uniform Linear Array (ULA) in various underwater scenarios [23]. An et al. proposed a combination of a linear array composed of multiple mutually perpendicular sub-arrays, overcoming the ambiguity of a single linear array's port and starboard orientation [24]. Under normal circumstances, both Ahmed and An had achieved research results, but in unconventional situations, such as pulse environments, it is worth exploring the advanced nature of the algorithms.

In recent years, there has been significant development in DOA estimation algorithms based on SBL. SBL is a statistical inference technique that is used to estimate sparse signals from noisy and incomplete data. It is a type of Bayesian regularization method that aims to find the most probable solution to an inverse problem by incorporating prior knowledge and assumptions about the underlying signal. In the context of DOA estimation, SBL is used to estimate the sparse signal of the DOA parameters from the array measurements. The key idea of SBL is to formulate the DOA estimation problem as a Bayesian inference problem, where the unknown DOA parameters are modeled as random variables, and the prior distribution of the DOA parameters is assumed to be sparse. By incorporating the prior information about the sparsity of the DOA parameters, SBL can effectively suppress the noise and interference in the array measurements and accurately estimate the DOA parameters, even in the presence of a limited number of snapshots. SBL algorithms typically involve iterative optimization procedures that update the estimates of the unknown parameters based on the observed data and the prior distribution. These algorithms can be computationally intensive, but they have been shown to be effective in a wide range of DOA estimation applications, including radar, sonar, and wireless communications.

Wang et al. aimed at the problem of interactions among the hydrophone array elements of the actual sonar array, which causes estimation performance dropping of the array's DOA, and a DOA estimation method under uncertain interactions of the array elements was proposed. However, the author did not note the relevant signals [25]. In order to achieve the high-precision Direction-of-Arrival (DOA) estimation of array signals in complex underwater acoustic environments, the root off-grid sparse Bayesian learning (ROGSBL) algorithm was applied to an underwater acoustics field [26]. In 2022, Haodong Bai studied the efficient DOA processing algorithm under multi-snapshots by aiming at the problem that the DOA estimation method, based on sparse Bayesian learning under single snapshots, has a low estimation accuracy and a large number of operations for increasing the number of snapshots [27]. He et al. proposed the SS-OGSBI algorithm to solve the problem of off-grid DOA estimation under coherent sources [28]. Guo et al. applied sparse Bayesian learning to the DOA estimation of underdetermined broadband signals with mutual arrays in unknown noise fields [29], while Shen et al. proposed an off-grid DOA estimation method based on subspace fitting and block-SBL to address the poor performance of traditional SBL-based DOA estimation algorithms under low SNR conditions [30]. Other influential works in this field included Yu et al., Ma et al., Zhu et al., Jimenez-Martinez M and Zhang et al. [31–35].

Although researchers have provided answers to the questions raised and made significant contributions to the field of Direction-of-Arrival (DOA) estimation using beamforming in underwater acoustics, the algorithms themselves have limitations. The researchers conducted their studies under the background of Gaussian noise, and further investigation is necessary to determine the robustness of the algorithms in highly impulsive noise environments.

In this article, a high robustness underwater target estimation technique based on variational sparse Bayesian is put forward by studying and analyzing the sparse prior assumption characteristics of the signal. The method models the observed signal by modeling the pulse noise, completes the derivation of the conditional distribution of the observed variables and the prior distribution of the sparse signal, and then combines the approx-

imate posterior distribution obtained by the VB method to obtain the recovered sparse signal, thereby reducing the impact of pulse noise on the estimation system. Finally, the performance of the aforementioned method was validated through simulation experiments.

## 2. Materials and Methods

### 2.1. Uniform Linear Array Signal Model

The linear array model is a fundamental mathematical framework for addressing the problem of sound source direction estimation. The model posits the existence of a linear array composed of multiple small sound sources, each of which continuously emits the same sound wave signal. These sound waves propagate through distinct paths to reach the receiving array, where the signal measured by each receiving element is expressed as a weighted sum of the signals stemming from each emitting source. More specifically, the linear array model comprises a transmit array and a receive array. Each sound-emitting source within the transmit array emits identical sound wave signals, which subsequently arrive at different receiving elements in the receive array via various propagation paths. The signal measured by each receiving element in the receive array is then computed as a weighted sum of the signals originating from the sound-emitting sources. These weighting coefficients reflect the path delay and attenuation factors experienced by the sound wave signal as it travels from the emitting source to the receiving element. Through the processing of signals within the linear array model, the direction of the sound source can be estimated. This involves calculating key parameters, such as the time delay and phase difference between individual receiving elements within the receiving array. Hence, the linear array model finds extensive applications in fields such as sound source direction estimation, sound beamforming, and signal source separation.

Consider an M-element ULA, the observation vector of the array can be defined as:

$$\mathbf{y}(t) = \mathbf{A}(\theta)\mathbf{S}(t) + \mathbf{n}(t), \quad t = 1, 2, \ldots, T \tag{1}$$

Here, $\mathbf{n}(t)$ represents independent identically distributed Gaussian white noise. The array manifold matrix is denoted by $\mathbf{A}(\theta)$ and denoted as $\mathbf{A}(\theta) = [\mathbf{a}(\theta_1), \mathbf{a}(\theta_2), \cdots, \mathbf{a}(\theta_n)]$. The matrix $A$ of size $M \times N$ represents the phase information of the array, where $N$ is the number of signal sources, and $M$ is the number of sensors.

The covariance matrix for the array output is defined as follows:

$$\mathbf{R} = \widetilde{\mathbf{A}}(\theta)\mathbf{A}^{\mathbf{H}}(\theta) + \delta^2\mathbf{I} = \sum_{n=1}^{N} \mathbf{P}_n\mathbf{a}(\theta_n)\mathbf{a}^{\mathbf{H}}(\theta_n) + \delta^2\mathbf{I} \tag{2}$$

In this equation, $\mathbf{P}_n = \mathbf{E} = \left[\left|\widetilde{\mathbf{S}}_n(t)\right|^2\right]$.

In practical applications, the correlation matrix is commonly used to estimate the output covariance matrix of the array. The correlation matrix is represented as follows:

$$\widetilde{\mathbf{R}} = \frac{1}{T}\sum_{t=1}^{T} \mathbf{x}(t)\mathbf{x}^{\mathbf{H}}(t) \tag{3}$$

In order to explain the principle of ULA more clearly, it can be explained in more detail in Figure 1.

### 2.2. Pulse Noise Distribution Model—Student-t Distribution

In this section, we will present an exposition on the Student-t distribution from three perspectives: origin, definition, and frequency spectrum. Regarding the parameter settings of the noise model in the frequency spectrum section, we will adopt the parameters used in the experiment described in this article as the standard. The primary objective is to provide a more intuitive illustration of the advantages of replacing pulse noise with the Student-t distribution model.

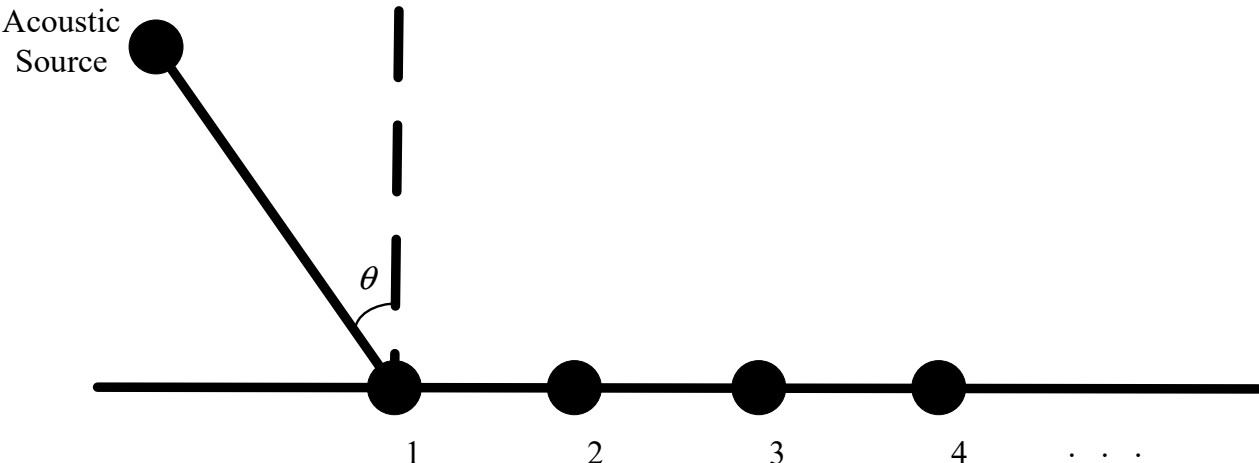

**Figure 1.** Uniform Linear Array signal model.

The main objective of array signal processing is to effectively remove noise from noisy observation data, thereby enabling the accurate recovery of the original signal and extraction of desired information. In many areas of array signal processing, narrowband signal models are commonly assumed, and noise is modeled as following a Gaussian distribution due to the fact that the Gaussian distribution satisfies the central limit theorem and has finite second-order and higher-order statistics. Additionally, signal characteristics can be represented by the mean and variance at any time. However, in real-world experimental environments, many types of noise do not adhere to a Gaussian distribution model. Such noises typically exhibit instantaneous pulse characteristics and more frequent abnormal data compared to Gaussian noise. Therefore, using a Gaussian distribution model to replace the noise model is not a realistic approach. For instance, if pulse noise is present in the DOA estimation environment, the noise distribution would have a heavy tail and a distribution with a heavier tail would be required to replace the Gaussian distribution.

Pulse noise models can be classified into two categories based on their generation mechanism: real physical statistical models and theoretical analytical models. Compared to physical statistical models, theoretical analytical models have relatively fixed mathematical expressions, which makes them more convenient for theoretical analysis. In the field of DOA estimation in the pulse environment, three models have been widely used, including the mixed Gaussian distribution model, the Alpha stable distribution model, and the Student-*t* distribution model. This paper primarily models pulse noise using the Student-*t* distribution, and the fundamental concepts of the Student-t distribution will be elaborated in detail below.

Gosset was a quality control officer at a brewery in 1908 when he discovered and proposed the Student-*t* distribution. At that time, he needed to study the variability of beer brewing in small sample sizes. However, since the data samples that he studied were very small, he could not use a traditional normal distribution for statistical analysis. To remedy this issue, Gosset examined the distribution of the population mean given the sample mean and sample standard deviation. He discovered that if the sample came from a normal distribution, the difference between the sample mean and the population mean could be described by a new distribution, which was later named the Student-t distribution. Gosset initially dubbed this distribution the "distribution of errors" because it was used to describe the error between the sample mean and the population mean. Later, the Student-t distribution became widely used in statistics, and it was named after Gosset's pen name, "Student". The Student-t distribution is a probability distribution that is commonly utilized to model data with heavy tails, i.e., tail probabilities that are significantly higher than those of a normal distribution.

The model of the Student-t distribution can be defined as follows:

$$n(t) \sim S(v|u, \mathbf{\Lambda}, \varsigma) \tag{4}$$

where $u$ is the average of the $M$-dimensional vector, $v$, $\mathbf{\Lambda} = \mathrm{diag}(\Lambda_1, \Lambda_2, \cdots, \Lambda_M)$ denotes the precision matrix, and $\varsigma$ denotes the degree of freedom (DOF) parameter. The decay becomes slower as the DOF decreases. When the degrees of freedom decrease, the shape of the Student-$t$ distribution changes, with the peak of the probability density function becoming lower and the tails becoming thicker. This makes it better suited for describing pulse noise. In order to explain the student's t-distribution more clearly, we did a simple simulation experiment and obtained the results shown in Figure 2.

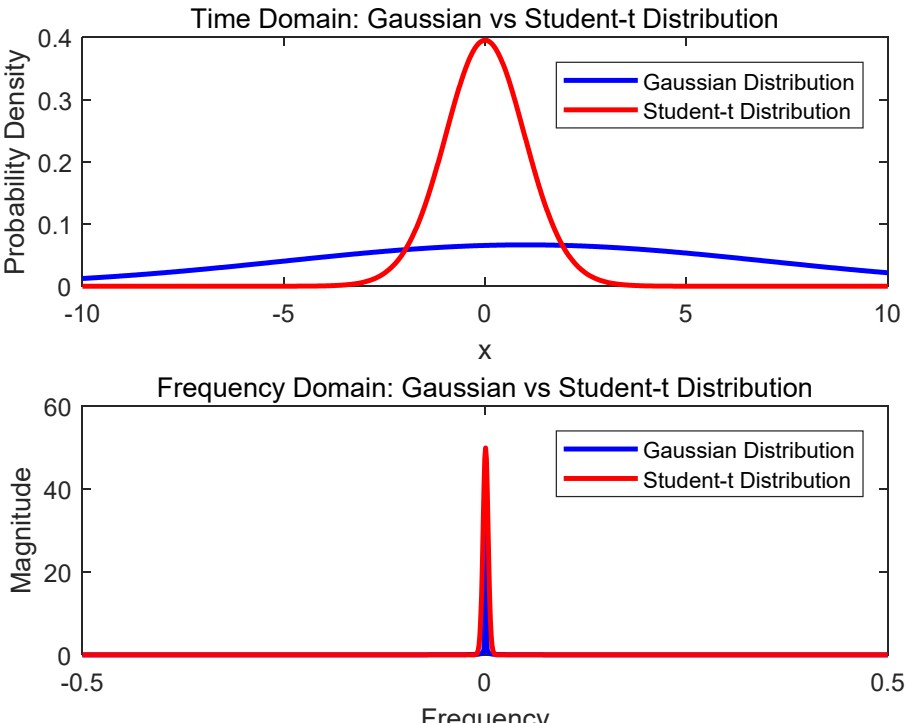

**Figure 2.** Student-$t$ distribution in the time-domain spectrum.

This experiment was mainly used to plot the time-domain and frequency-domain graphs of the Gaussian distribution and Student-$t$ distribution. We first set three parameters: mean = 1, standard deviation = 6, and degrees of freedom = 30. Then, it generated an $x$-axis vector containing 1000 points using the "linspace" function, used to represent the continuous variable, x, in the time-domain. Next, the probability density functions of the Gaussian distribution and Student-$t$ distribution were calculated to generate the time-domain graphs of the two distributions. The program used the "plot" function to plot the time-domain graphs of the two distributions.

The next part of the experiment was used to plot the frequency-domain graphs of the Gaussian distribution and Student-$t$ distribution. The experiment first used the "fft" function to calculate the Fourier transform of the time-domain graph and used the "fft-shift" function to center the result. Then, we used the "linspace" function to generate the continuous variable "freq" in the frequency domain and used the "plot" function to plot the frequency-domain graphs of the two distributions and added axis labels, legends, and titles.

When plotting the graphs, the program used the "hold on" function to make both graphs of the distributions plotted on the same figure. This is performed to better compare the differences between the two distributions.

In the present illustration, it can be observed that the time-domain spectrum of the Student-*t* distribution exhibits a shape similar to that of the Gaussian distribution. However, compared to the Gaussian distribution, under the parameters set in this experiment, the Student-*t* distribution is better able to model noise with local outliers, such as impulse noise. Additionally, in the frequency-domain spectrum, the frequency response of the Student-*t* distribution is smoother than that of the Gaussian distribution. That is, its amplitude changes more slowly with frequency, which also helps to reduce the impact of impulse noise in the high-frequency range. Therefore, in this paper, the Student-*t* distribution is adopted as the model for impulse noise.

*2.3. Graphical Models*

The interaction between entities involved in a probabilistic system is represented by a graphical model, where nodes represent random variables, and arrows depict dependencies between variables [36]. A directed arrow from node A to node B indicates that the value of random variable B depends on the value of random variable A. Graphical models can be categorized into directed graph models and undirected graph models [15,36]. This paper focuses on directed graph models, also known as Bayesian network graphical models [37].

The definition of a directed graph model is as follows:

Given the conditional probability distribution of each node in the graphical model, the formula for calculating the joint distribution over all variable sets is $p(x)$ [38].

$$p(x) = \prod_s p\left(x_s \middle| x_{\pi(s)}\right) \tag{5}$$

Figure 3 shows an example of a directed graph model. In this model, *a*, *b*, and *d* represent random variables, and each node in the graphical model represents a conditional probability density. If the probability density of the node is unknown, it can be parameterized by a set of parameters. The joint distribution of the probability density is then expressed as follows:

$$p(a, b, d) = p(a; \theta_1) p(b; \theta_2) p(d|a, b; \theta_3) \tag{6}$$

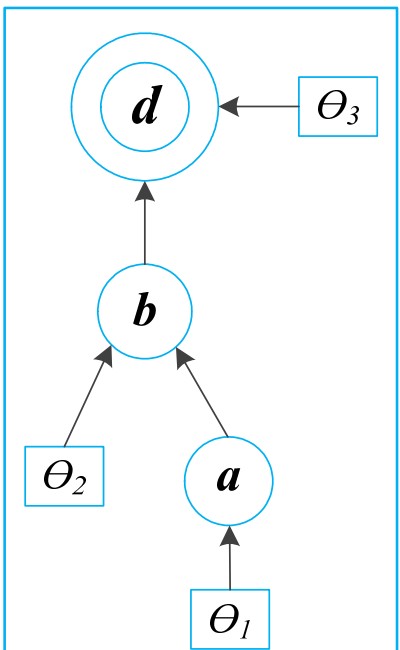

**Figure 3.** Example of a directed graph model.

The above expression can be simplified by considering the independence implied by the structure of the graphical model. Generally speaking, in the graphical model, each node is independent of its higher-level nodes. Therefore, expression (6) can be simplified as follows:

$$p(a, b, d) = p(a; \theta_1) p(b; \theta_2) p(d|a; \theta_3) \tag{7}$$

Another function of the graphical model is to arbitrarily distinguish random variables into those with directly observed results and those with hidden random variables without directly observed results [39]. In addition, the graphical model can be divided into parameterized graphical models and non-parameterized graphical models. If it is a parameterized graphical model, the parameters will appear in the conditional probability distribution of some nodes; that is, the probability models of these distributions are parameterized probability models.

## 3. Theoretical Model

In this paper, a high robustness underwater target estimation method based on variational sparse Bayesian inference is proposed by studying and analyzing the sparse prior assumption characteristics of the signal. The method models the observed signal by modeling the pulse noise, completes the derivation of the conditional distribution of the observed variables and the prior distribution of the sparse signal, and then combines the VB method to obtain the approximate posterior distribution, thereby obtaining the recovered signal of the sparse signal.

Firstly, it is assumed that there exist N narrowband signals impinging upon an M-element linear array.

$$\mathbf{Y} = \mathbf{\Phi}\mathbf{X} + \mathbf{N} \tag{8}$$

In the equation, the observation matrix is represented by $\mathbf{Y} \in \mathbb{C}^{M \times L}$, $\mathbf{X} \in \mathbb{C}^{N \times L}$ represents the original signal, $\mathbf{N} \in \mathbb{C}^{M \times L}$ represents the noise matrix, and $\mathbf{\Phi} \in \mathbb{C}^{M \times N}$ represents the measurement matrix.

In the above Bayesian model, the joint distribution of all the observed variables and unknown variables is required, which usually includes the conditional distribution of the observed variables and the prior distribution of the sparse signal. The conditional distribution of the observed variables and the prior distribution of the sparse signal are derived, and the approximate posterior distribution is obtained through the VB method, thereby obtaining the recovered signal of the sparse signal.

### 3.1. Derivation of Conditional Distribution of Observation Variables

Modeling the pulse noise using the Student-*t* distribution, the probability density function is given as follows:

$$S(v|u, \Lambda, \varsigma) = \frac{\Gamma\left(\frac{M+\varsigma}{2}\right)}{\Gamma\left(\frac{\varsigma}{2}\right)(\varsigma\pi)^{\frac{M}{2}}} |\Lambda|^{\frac{1}{2}} \left[1 + \frac{(v-u)^{\mathrm{T}}\Lambda(v-u)}{\varsigma}\right]^{-\frac{M+\varsigma}{2}} \tag{9}$$

where $\Gamma(\cdot)$ denotes the Gamma function, assuming that all the columns of the noise matrix are independent and follow a zero-mean Student-*t* distribution [40].

By introducing the latent variable, $\lambda$, the Student-*t* distribution is an infinite mixture of Gaussian distributions with variances extended by gamma distributions [41].

$$S(\mathbf{v}|\mathbf{u}, \mathbf{\Lambda}, \zeta) = \int_0^{\infty} \mathrm{N}\left(\mathbf{v}\Big|\mathbf{u}, (\lambda\mathbf{\Lambda})^{-1}\right) \mathrm{G}(\lambda|\zeta/2, \zeta/2) d\lambda \tag{10}$$

Here, $\mathrm{N}(\cdot)$ And $\mathrm{G}(\cdot)$ are the Gaussian distribution and Gamma distribution, respectively.

Therefore, the conditional distribution of the observed variables can be written as follows:

$$\begin{cases} p(\mathbf{Y}|\mathbf{X}, \boldsymbol{\Lambda}, \zeta) = \prod_{l=1}^{L} \mathrm{N}\left(\mathbf{y}_l \middle| \boldsymbol{\Phi}_l (\lambda \boldsymbol{\Lambda})^{-1}\right) \\ p(\lambda|\zeta) \neq \mathrm{G}(\lambda|\zeta/2, \zeta/2) \end{cases} \tag{11}$$

By placing the Gamma distributions on each diagonal element of $\zeta$ and $\boldsymbol{\Lambda}$, these equations can be obtained:

$$p(\zeta) = \mathrm{G}(\zeta|c, d) \tag{12}$$

$$p(\boldsymbol{\Lambda}) = \prod_{l=1}^{L} \mathrm{G}(\Lambda_m|a_m, b_m) \tag{13}$$

In the equation, $a_m$, $b_m$, $c$, and $d$ are hyperparameters of the Gamma distribution.

### 3.2. Derivation of Sparse Signal Prior Distribution

Assuming that all rows of the matrix, $\mathbf{X}$, are independent and follow a Gaussian distribution, the prior distribution of the sparse signal, s, can be obtained [41].

$$p(\mathbf{X}|\boldsymbol{\gamma}) = \prod_{n=1}^{N} \mathrm{N}\left(\mathbf{x}_{n,\cdot} \middle| 0, \gamma_n^{-1} \mathbf{I}_{L \times L}\right) \tag{14}$$

Here, $\boldsymbol{\gamma} = [\gamma_1, \cdots, \gamma_n]$ represents the precision vector of the sparse signal, $\mathbf{X}$, and a Gamma distribution with hyperparameters $\partial_n$ and $\beta_n$ are used for each precision vector, $\gamma_n$, as follows [41,42]:

$$p(\gamma) = \prod_{n=1}^{N} \mathrm{G}(\gamma_n|\partial_n, \beta) \tag{15}$$

According to Equations (12)–(15), the joint distribution of all the observed variables and unknown variables can be decomposed as follows:

$$\begin{aligned} p(\mathbf{Y}, \mathbf{X}, \gamma, \boldsymbol{\Lambda}, \zeta, \lambda) &= p(\mathbf{Y}|\mathbf{X}, , \boldsymbol{\Lambda}, \lambda) p(\mathbf{X}|\gamma) p(\boldsymbol{\Lambda}) p(\lambda|\zeta) p(\zeta) \\ &= \left(\prod_{l=1}^{L} p(\mathbf{y}_l|\mathbf{x}_l, \boldsymbol{\Lambda}, \lambda)\right) \left(\prod_{n=1}^{N} p(\mathbf{x}_n|\gamma_n) p(\gamma_n)\right) \times \\ &\quad \left(\prod_{m=1}^{M} p(\Lambda_m)\right) p(\lambda|\zeta) p(\zeta) \end{aligned} \tag{16}$$

In Figure 4, more detailed dependencies can be obtained about variables and unknown variables.

### 3.3. Variational Bayes

Bayesian inference is based on the posterior distribution, $p(\boldsymbol{\Omega}|\mathbf{Y}) = p(\mathbf{Y}, \boldsymbol{\Omega})/p(\mathbf{Y})$, where $\boldsymbol{\Omega}$ represents the set of all unknown variables. However, because the marginal distribution, $p(\mathbf{Y})$, can be difficult to handle, Bayesian inference often requires approximation. In the VB method, an approximation of $p(\boldsymbol{\Omega}|\mathbf{Y})$, is made using a coefficient distribution, $q(\boldsymbol{\Omega}) = q(\mathbf{X})q(\gamma)q(\boldsymbol{\Lambda})q(\zeta)q(\lambda)$, where $\boldsymbol{\Omega} = \{\mathbf{X}, \boldsymbol{\Lambda}, \gamma, \lambda, \zeta\}$, and each approximate distribution in $q(\boldsymbol{\Omega})$ is obtained by computing the logarithmic expectation of (16) with respect to other distributions.

$$\ln q(\mathbf{X}) \propto \sum_{l=1}^{L} \left\{ \begin{array}{l} -\frac{1}{2}\mathbf{x}_l^T \left(\mathrm{E}[\lambda]\mathrm{E}[\boldsymbol{\Lambda}] + \boldsymbol{\Phi}^T \mathrm{diag}\mathrm{E}[\gamma]\boldsymbol{\Phi}\right)x_l + \\ \frac{1}{2}\left(\boldsymbol{\Phi}^T \mathrm{diag}\mathrm{E}[\gamma]\mathbf{y}_l\right)^T + \frac{1}{2}\mathbf{x}_l^T \left(\boldsymbol{\Phi}^T \mathrm{diag}\mathrm{E}[\gamma]\mathbf{y}_l\right) \end{array} \right\} \tag{17}$$

Here, $\mathrm{E}[\cdot]$ denotes the expectation operator. By Equation (17), $q(\mathbf{X})$ can be calculated as follows:

$$q(\mathbf{X}) = \prod_{l=1}^{L} \mathrm{N}(\boldsymbol{\mu}_l, \boldsymbol{\Sigma}) \tag{18}$$

where $\boldsymbol{\mu}_l$ represents the expectation and $\boldsymbol{\Sigma}$ is the variance [43].

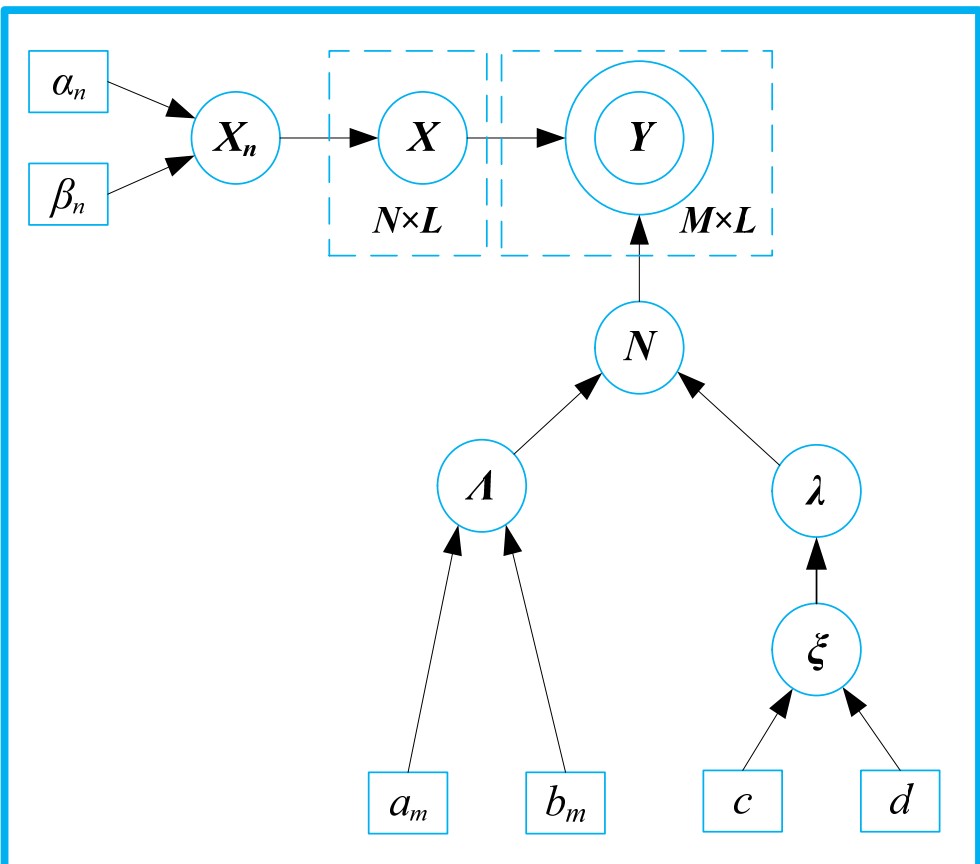

**Figure 4.** Graphical model.

$$\begin{cases} \mathbf{\Sigma} = \left(\mathbf{A} + \mathbf{\Phi}^T \mathbf{B} \mathbf{\Phi}\right)^{-1} \\ \mathbf{\mu}_l = \mathbf{\Sigma} \mathbf{\Phi}^T \mathbf{B} \mathbf{y}_l \ , \qquad l = 1, 2, \cdots, L \end{cases} \tag{19}$$

Similarly, $q(\gamma)$, $q(\mathbf{\Lambda})$, $q(\zeta)$, and $q(\lambda)$ is obtained one by one and expressed in the following form:

$$\begin{aligned} q(\zeta) &= \mathbf{G}(\zeta | c^*, d^*) \\ q(\lambda) &= \mathbf{G}(\lambda | \zeta_1^*, \zeta_2^*) \\ q(\gamma) &= \prod_{n=1}^{N} \mathbf{G}(\gamma_n | \alpha_n^*, \beta_n^*) \\ q(\mathbf{\Lambda}) &= \prod_{m=1}^{M} \mathbf{G}(\Lambda_m | a_m^*, b_m^*) \end{aligned} \tag{20}$$

The approximated hyperparameters are defined as follows [44,45]:

$$\begin{cases} c^* = c + \frac{1}{2} \\ d^* = d + \frac{1}{2}(\mathrm{E}[\lambda] - \mathrm{E}[\ln \lambda] - 1) \end{cases} \tag{21}$$

$$\begin{cases} \zeta_1^* = \frac{1}{2}(\mathrm{E}[\zeta] + M) \\ \zeta_2^* = \frac{1}{2} \sum_{m=1}^{M} \left\{ \mathrm{E}[\Lambda_m] \sum_{l=1}^{L} \mathrm{E}\left[(\mathbf{y}_l - \phi_m \mathbf{x}_l)^2\right] \right\} + \frac{1}{2}\mathrm{E}[\zeta] \end{cases} \tag{22}$$

$$\begin{cases} a_m^* = a_m + \frac{1}{2} \\ b_m^* = b_m + \frac{1}{2}\mathrm{E}[\lambda] \sum_{l=1}^{L} \mathrm{E}\left[(\mathbf{y}_l - \phi_m \mathbf{x}_l)^2\right] \end{cases} \tag{23}$$

$$\begin{cases} \alpha_n^* = \alpha_n + \frac{1}{2} \\ \beta_n^* = \beta_n + \frac{1}{2}\mathrm{E}\left[\mathbf{x}_n \mathbf{x}_n^T\right] \end{cases} \tag{24}$$

Based on the approximate distribution, (17), and Equation (20), the expectations of the approximated hyperparameters are expressed as follows [45]:

$$
\begin{aligned}
\mathrm{E}[\mathbf{x}_l] &= \boldsymbol{\mu}_l \\
\mathrm{E}\left[\mathbf{x}_n \mathbf{x}_n^{\mathrm{T}}\right] &= \left( \sum_{l=1}^{L} \left( \boldsymbol{\mu}_l \boldsymbol{\mu}_l^{\mathrm{T}} + \boldsymbol{\Sigma} \right) \right)_{nn}
\end{aligned}
\tag{25}
$$

$$
\begin{aligned}
\mathrm{E}[\lambda] &= \zeta_1^* / \zeta_2^* \\
\mathrm{E}[\ln \lambda] &= \varphi\left(\zeta_1^*\right) - \ln \zeta_2^*
\end{aligned}
\tag{26}
$$

$$
\begin{aligned}
\mathrm{E}[\Lambda_m] &= a_m^* / b_m^* \\
\mathrm{E}[\gamma_n] &= \alpha_n^* / \beta_n^* \\
\mathrm{E}[\zeta] &= c^* / d^*
\end{aligned}
\tag{27}
$$

$$
\mathrm{E}\left[(\mathbf{y}_l - \phi_m \mathbf{x}_l)^2\right] = \mathbf{y}_l^2 - 2\mathbf{y}_l \phi_m \mathrm{E}[\mathbf{x}_l] + \phi_m \left( \boldsymbol{\mu}_l \boldsymbol{\mu}_l^{\mathrm{T}} + \boldsymbol{\Sigma} \right) \phi_m^{\mathrm{T}}
\tag{28}
$$

Here, $\varphi(\cdot)$ represents the digamma function, and $\boldsymbol{\mu}_l$ represents the nth element on the main diagonal of the matrix.

Moreover, from Equation (25), the mean $\boldsymbol{\mu}_l$ gives an estimate of $\mathbf{x}_l$, and thus the recovery result of the sparse signal can be obtained by the following equation:

$$
\hat{\mathbf{X}} = [\boldsymbol{\mu}_1, \cdots, \boldsymbol{\mu}_l]
\tag{29}
$$

where the expectation can be obtained by coupling the hyperparameters in (21)–(24).

Therefore, the solution can be obtained by iteratively computing (19) and (21)–(24) until convergence, leading to the optimal recovery result.

To summarize, the proposed algorithm is listed in Table 1.

**Table 1.** Flow of target azimuth estimation algorithm based on variational sparse Bayes.

| Input | Observed Variables, s, and Measurement Matrix, *X* |
|:---:|:---:|
| 1 | Initialize hyperparameters $\{\alpha_n, \beta_n\}_{m=1}^{M}$, $\{\alpha_n, \beta_n\}_{n=1}^{N}$, $c$, and $d$, set stopping threshold, $\varepsilon$, and maximum iteration number, $I_{\max}$. |
| 2 | Calculate the mean and variance using Equation (19). |
| 3 | Update hyperparameters $\{a_m, b_m\}_{m=1}^{M}$, $\{\alpha_n^*, \beta_n^*\}_{n=1}^{N}$, $c$, and $d$ separately using Equations (21)–(24). |
| 4 | If the maximum change of hyperparameters is less than the stopping threshold, stop iterating and go to step 5. Otherwise, go back to step 2 to continue iterating. |
| 5 | Output the recovery result. |

## 4. Simulation Results

Consider a ULA with M = 30 elements spaced at half-wavelength. Three incoherent signal sources are assumed to be located in the far-field of the receiving array, an incident from the directions of 45°, 60°, and 90°, with an SNR of 5 dB. The iteration number is set to 500, and the number of grid points is set to N = 181. The measurement noise is generated using $f = (1 - p)N(0, \delta^2) + pN(0, k\delta^2)$ [46], where $N(0, \delta^2)$ represents the background noise, $N(0, k\delta^2)$ represents the pulse noise, $p$ represents the percentage of pulse noise, $k$ represents the intensity of the pulse noise, and $\delta^2$ represents the variance of the background noise. The parameters are set to $p = 0.3$, $k = 30$, and $\delta^2 = 1$. The initial parameters are set to $10^{-6}$ [47,48].

This section may be divided by subheadings. It should provide a concise and precise description of the experimental results, their interpretation, and the experimental conclusions that can be drawn.

### 4.1. Example 1: Single Snapshot Case

Figures 5 and 6 show the underwater DOA estimation results based on the improved SBL algorithm in both non-impulsive and impulsive noise environments. As can be seen from the figures, under the single snapshot condition, SBL can recover the signal well, regardless of whether it is in a non-impulsive or impulsive noise environment. The recovery result in the non-impulsive environment is better than that in the impulsive noise environment. From Figures 5d and 6d, it can be observed that the error between the two is not very large. Therefore, under the single snapshot condition, the algorithm has good estimation performance in the impulsive noise environment.

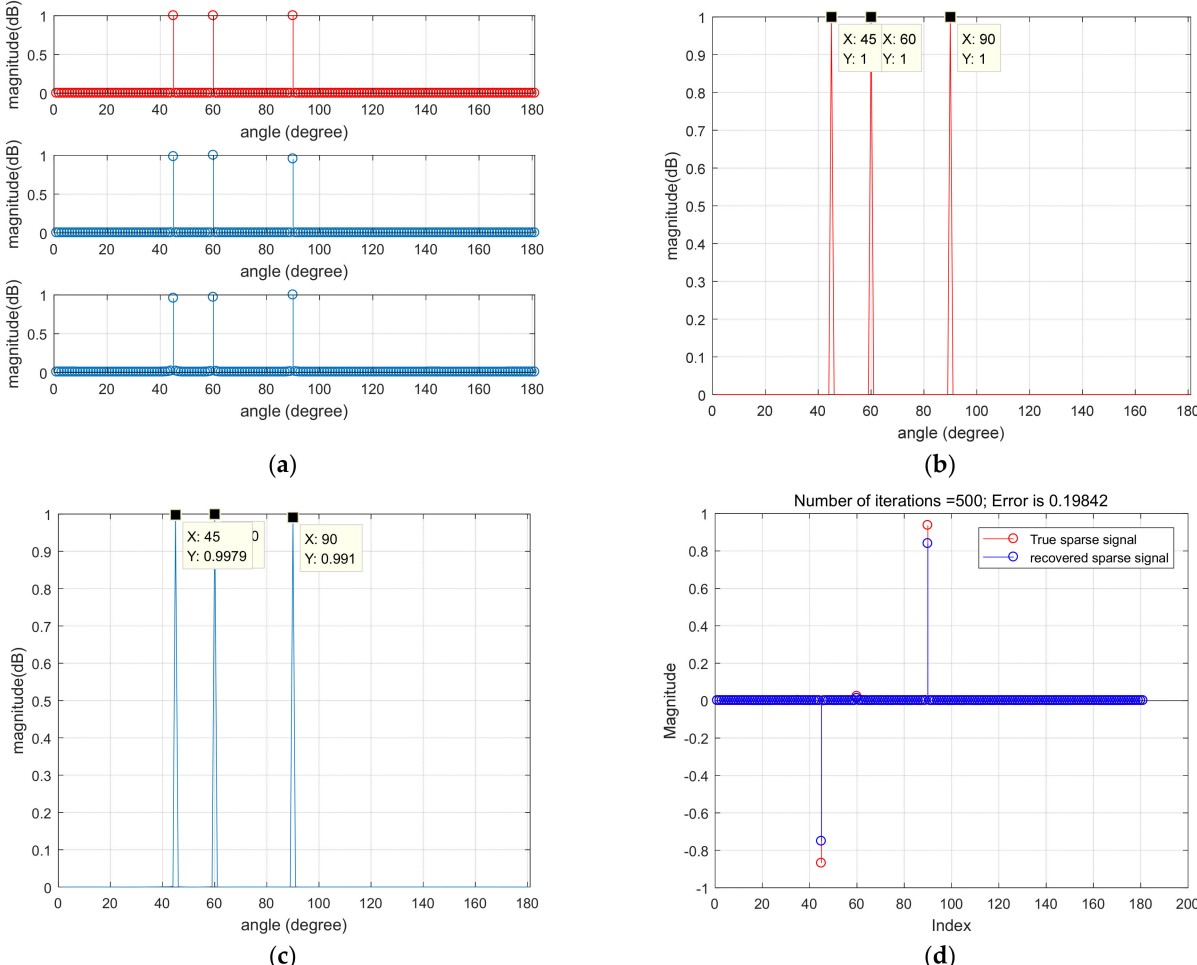

**Figure 5.** DOA estimation results of single snapshot in a non-pulsed environment. (**a**) Comparison of azimuth between the original signal and recovered signal; (**b**) bearing information of the original signal; (**c**) restoring the orientation information of the signal; (**d**) difference between estimated DOA and true DOA.

### 4.2. Example 2: Multiple Snapshot Case

In this section, the DOA estimation problem based on the SBL algorithm in the pulse noise environment under the multiple snapshot case is considered. The estimation results under Gaussian white noise are used as a reference to explore the estimation performance of the algorithm. The number of snapshots is set to 500, the frequency is $f = 1000$, and the sampling frequency is $f_s = 10 \times f$.

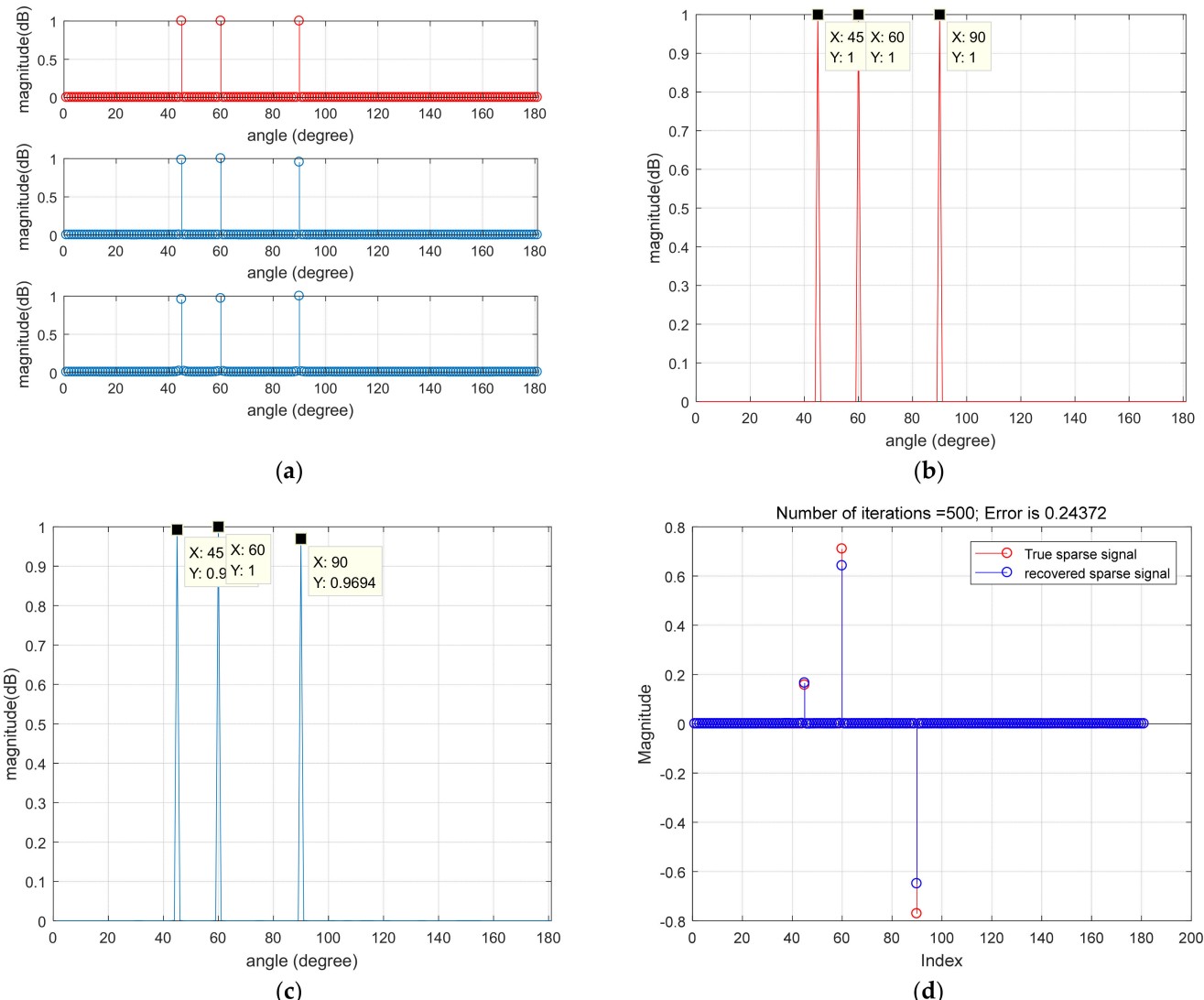

**Figure 6.** DOA estimation results of single snapshot under pulse environment. (**a**) Comparison of azimuth between the original signal and recovered signal; (**b**) bearing information of the original signal; (**c**) restoring the orientation information of the signal; (**d**) difference between estimated DOA and true DOA.

Figures 7 and 8 show the underwater DOA estimation results based on the improved variational Bayesian algorithm in both non-impulsive and impulsive noise environments under the multi-snapshot condition. As can be seen from the figures, the recovery results in the non-impulsive environment are better than those in the impulsive noise environment, but the errors between the recovered signal and the original signal are not large in either environment, which is consistent with theoretical expectations. On the other hand, from Figures 7d and 8d, as well as Figures 6d and 7d, it can be observed that the error between the recovered signal and the original signal under the single snapshot condition is larger than that under the multi-snapshot condition.

### 4.3. Example 3: The Comparison of Algorithm Performance

In this section, the root mean square error (RMSE) of the performance evaluation metric is introduced, and we analyzed the classic algorithms, CBF, MUSIC, and the proposed method. The basic experimental conditions were consistent with the first two experiments, with a snapshot number of 600, Monte Carlo iterations of 500, and an SNR ranging from −10 to 20. The results are shown in Figure 9.

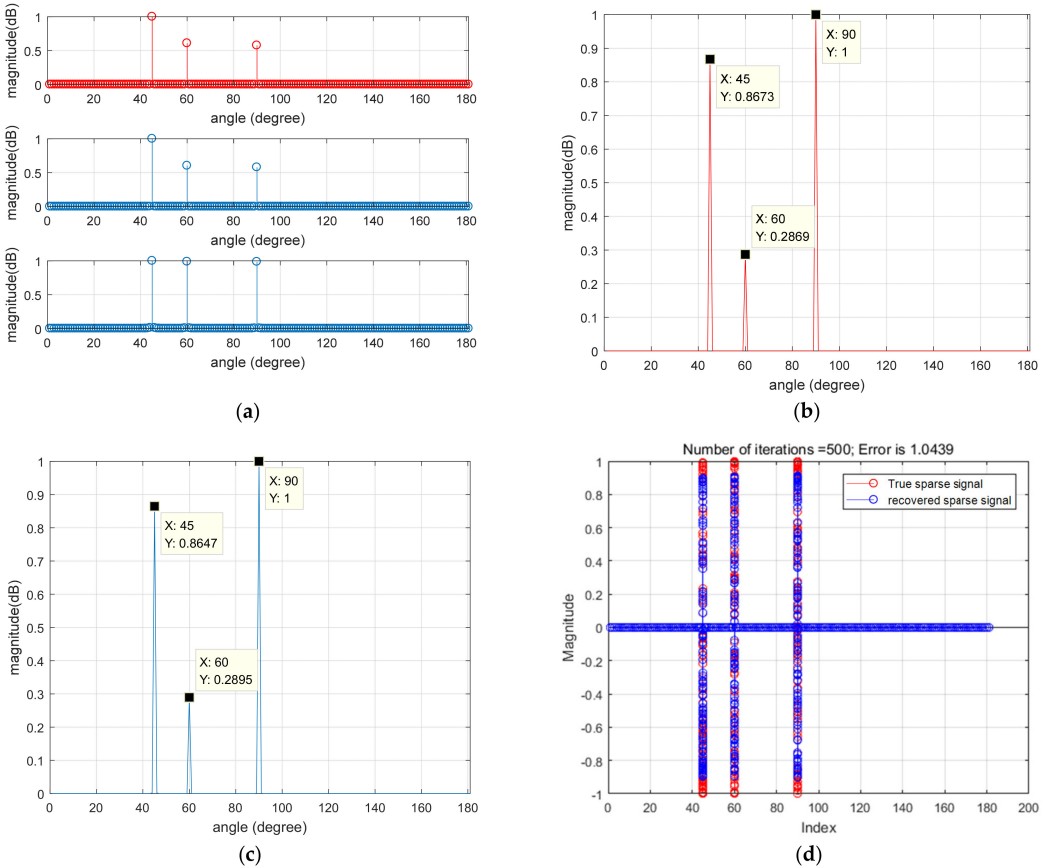

**Figure 7.** DOA estimation results of multi-snapshot in a non-pulsed environment. (**a**) Comparison of azimuth between the original signal and recovered signal; (**b**) bearing information of the original signal; (**c**) restoring the orientation information of the signal; (**d**) difference between estimated DOA and true DOA.

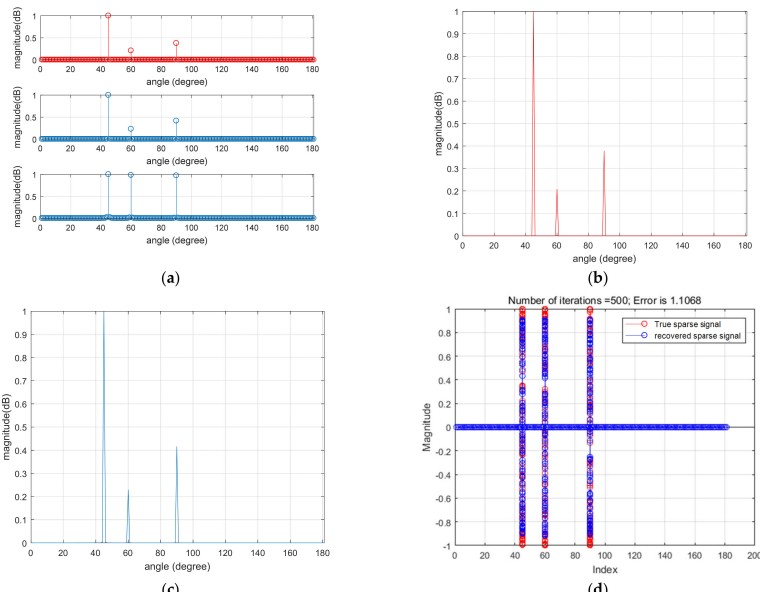

**Figure 8.** DOA estimation results of multi-fast in pulsed environment. (**a**) Comparison of azimuth between the original signal and recovered signal; (**b**) bearing information of the original signal; (**c**) restoring the orientation information of the signal; (**d**) difference between estimated DOA and true DOA.

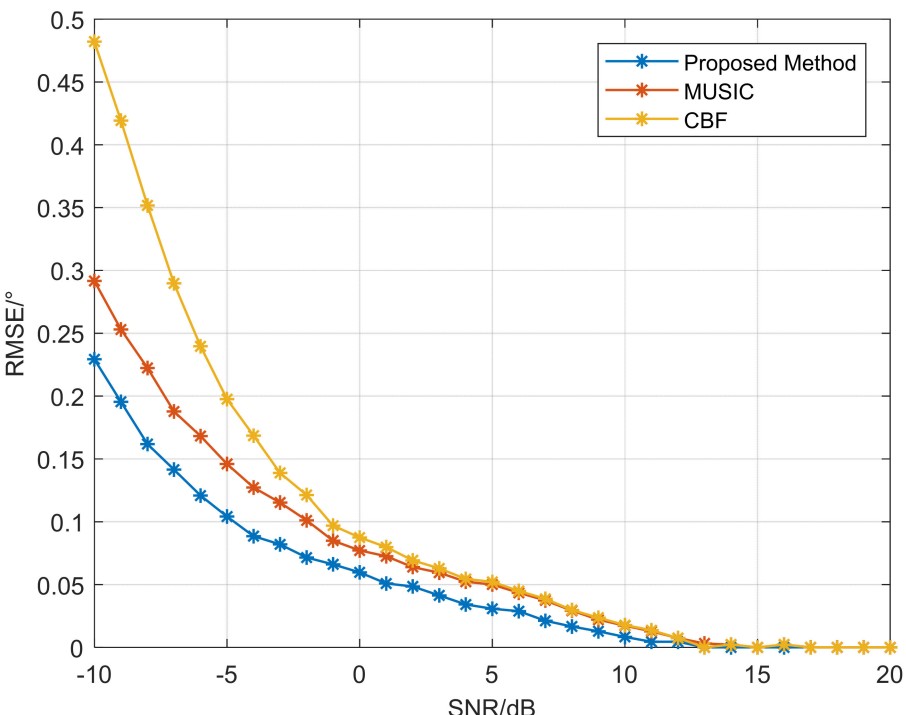

**Figure 9.** The comparison of algorithm performance.

The RMSE is a metric used to measure the difference between predicted and actual values of a variable. It is computed by taking the square root of the average of the squared differences between the predicted and actual values. The RMSE is commonly employed to evaluate the accuracy of predictive models or estimation methods, where a lower value indicates better performance. The formula for calculating RMSE is as follows:

$$\text{RMSE} = \sqrt{\frac{1}{PK}\sum_{p=1}^{P}\sum_{k=1}^{K}\left(\hat{\theta}_{p,n} - \theta_k\right)^2} \tag{30}$$

where $K$ represents the number of signal sources, $P$ represents the number of Monte Carlo experiments, and $\theta_k$ represents the estimated value of the Kth target angle in the Pth Monte Carlo experiment.

Figure 9 illustrates how the RMSE varies for different algorithms at different SNRs. As depicted in the figure, the RMSE curves for CBF, MUSIC, and the proposed method all decrease as the SNR increases. Furthermore, when the SNR is very high, the RMSE for all three algorithms is significantly small. Notably, the RMSE curve for the proposed method remains consistently lower than those of the other algorithms, implying that it has superior estimation performance.

## 5. Discussion

Despite notable advances in array signal processing, the increasing complexity and diversity of signal environments pose challenges to the conventional narrowband signal-based DOA estimation algorithm. The method focuses on recovering underwater DOA estimation signals under pulse interference and presents a novel underwater target azimuth estimation algorithm based on SBL, which overcomes the limitations of traditional narrowband signal-based DOA estimation algorithms in complex signal environments. The simulation results demonstrate that the algorithm accurately recovers the source signal in both single and multi-snapshot scenarios. The algorithm's performance was evaluated using the RMSE, which demonstrated that the algorithm outperforms other algorithms in signal recovery, effectively addressing the problem of low estimation accuracy in pulse

environments. The algorithm can be applied in underwater target tracking and localization. Future work will explore azimuth estimation and signal recovery under wideband signals.

**Author Contributions:** This research article was a collaborative effort that involved the contributions of several authors. L.D. and H.L. contributed equally to the work and are listed as the first and second authors, respectively. Z.L. served as the corresponding author and provided guidance throughout the study. H.L. was responsible for designing and conducting the experiments, analyzing the data, and writing the manuscript. Specifically, H.L. developed the research question and hypotheses, designed the study protocol, recruited participants, collected and managed the data, conducted statistical analyses, and interpreted the findings. H.L. also wrote the initial draft of the manuscript and revised it based on feedback from the other authors. L.D. and L.W. assisted with the experimental design, performed data analysis, and helped to revise the manuscript. Specifically, L.D. contributed to the development of the study protocol, assisted with data collection and management, conducted statistical analyses, and L.W. helped to interpret the findings. L.D. and L.W. also provided critical feedback on the manuscript and helped to revise it. Z.L. provided overall guidance throughout the study and served as the corresponding author. Specifically, Z.L. helped to conceive the study, provided input on the research question and hypotheses, supervised the experimental design and data collection, and provided guidance on the statistical analyses and interpretation of the findings. Z.L. and X.L. played key roles in revising and finalizing the manuscript. Each author has made significant contributions to the research and preparation of the manuscript. The contributions of each author reflect their individual expertise and skills and demonstrate their commitment to advancing scientific knowledge in their respective fields. In conclusion, this research article was a collaborative effort that involved the contributions of several authors. H.L. and L.D. contributed equally to the work, respectively. Z.L. served as the corresponding author and provided guidance throughout the study. Together, the authors developed the research question and hypotheses, designed the study protocol, collected and analyzed the data, and interpreted the findings. The contributions of each author were critical to the success of the study and demonstrate their commitment to advancing scientific knowledge in their respective fields. All authors have read and agreed to the published version of the manuscript.

**Funding:** This research was supported by several funding sources, including the Shandong Province "Double-Hundred" Talent Plan (WST2020002), Key R&D programs (2022YFC2808003; 2023YFE0201900), and the Open Project of the State Key Laboratory of Sound Field Acoustic Information (No. SKLA202203).

**Data Availability Statement:** Not applicable.

**Conflicts of Interest:** The authors declare no conflict of interest.

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
