# Peer review of "Research on High Robustness Underwater Target Estimation Method Based on Variational Sparse Bayesian Inference"

_remotesensing, doi:10.3390/rs15133222_

Round 1

Reviewer 1 Report (Previous Reviewer 3)

The paper proposes a high robustness underwater target estimation method based on variational sparse Bayesian inference by studying and analyzing the sparse prior assumption characteristics of signals, however, is necessary to clarify some process used to perform this analysis. Also, some factual clarifications will be helpful as highlighted in the comments below. 

Previous recommendations have been done (remotesensing-2427828), however, there are some of them that still need to be amended in order to understand the responses (signals) analysis.

-Page 2, lines 256-257. Section 2 deals with narrowband signals, how are evaluated all the bandwidth signals? And what is the effect of random process? 

-The decay becomes slower as the DOF decreases. How is related with the proposal?.

Author Response

Reply to Reviewer 1 of Manuscript

Firstly, we would like to give our thanks to Reviewer 1 for his patience on reviewing and the valuable comments on this manuscript. We have studied the comments carefully and give our explanation as follows which, we hope, would meet his satisfaction. The amendments highlighted in the revision manuscript. The paper has been polished by the native speaker. We also modified some errors in the revised manuscript.

Reviewer 1’s Comments:

  1. Page 2, lines 256-257. Section 2 deals with narrowband signals, how are evaluated all the bandwidth signals? And what is the effect of random process?
  2. The decay becomes slower as the DOF decreases. How is related with the proposal?

Our explanations:

Response 1:

First of all, we would like to express our gratitude to the reviewers for their careful review and valuable comments. To address the issues raised, we will clarify and supplement the relevant content in the revised manuscript. Here is our response to your questions:

Regarding the evaluation of broadband signals: In Section 2, we mainly focus on narrowband signals. The method proposed in this paper can also be applied to the evaluation of broadband signals, but some modifications and extensions are needed. In future work, we will continue to investigate the problems of direction estimation and signal recovery under broadband signals, and some explanations are provided at the end of the paper.

Impact of random processes: In signal processing, researchers often assume that the noise of the received signal is additive white Gaussian noise. However, in practical engineering, various random effects (such as multipath propagation, atmospheric turbulence, and unconventional noise) can introduce variability and disturb the signal, which may seriously affect the accuracy and precision of DOA estimation, and even lead to the failure of estimation methods. After extensive literature research, it is found that pulse noise has a great impact on conventional target direction estimation algorithms, and it may even cause the algorithm to fail. Therefore, we conducted a study on high robustness underwater target estimation method based on variational sparse Bayesian inference, aiming to avoid the excessive influence of pulse noise on DOA estimation.

To address the impact of random processes on DOA estimation, it is necessary to model the statistical characteristics of the signal and incorporate them into the DOA estimation algorithm. One approach is to use statistical signal processing techniques, such as maximum likelihood estimation, maximum a posteriori estimation, and Bayesian inference, which can estimate the statistical parameters of the signal and provide robust DOA estimation. In addition, developing adaptive algorithms is also a very promising direction. Adaptive algorithms can automatically adjust algorithm parameters based on the statistical characteristics of the received signal, thereby providing more accurate and reliable DOA estimation. This approach is particularly suitable for complex underwater environments, where the complexity and randomness of the underwater environment can cause the signals received by sensors to vary greatly.

In this paper, we model the signal using statistical signal processing techniques and use Bayesian inference to estimate the signal parameters, thereby obtaining more reliable DOA estimation results.

In summary, the impact of random processes on the DOA estimation method based on variational Bayesian inference is significant. Therefore, it is necessary to consider the impact of these processes and develop methods that are robust to changes in the received signal.

Response 2:

Thank you for your feedback on our manuscript. The issue of slower decay as the degrees of freedom decrease is closely related to our proposed research. This is because when the degrees of freedom decrease, the shape of the Student's t-distribution changes, with the peak of the probability density function becoming lower and the tails becoming thicker. This makes it better suited for describing pulse noise, which is one of the reasons why we chose the Student's t-distribution for pulse modeling. In lines 264, 287 and 296 of the article, we provided corresponding explanations and highlighted them with special colors.

In the revised manuscript, we have provided a detailed explanation of this relationship and how it relates to our proposed research. We apologize for any confusion caused by the brief description in the previous version. Once again, thank you for your valuable feedback.

Line 264 – Line 266: For instance, if pulse noise is present in the DOA estimation environment, the noise distribution would have a heavy tail, and a distribution with a heavier tail would be required to replace the Gaussian distribution.

Line 287 – Line 289: The Student-t distribution is a probability distribution that is commonly utilized to model data with heavy tails, i.e., tail probabilities that are significantly higher than those of a normal distribution.

Line 293 – Line 296: When the degrees of freedom decrease, the shape of the Student's t-distribution changes, with the peak of the probability density function becoming lower and the tails becoming thicker. This makes it better suited for describing pulse noise.

Reviewer 2 Report (New Reviewer)

1.      In the paper, the authors clearly explain the research contribution and innovation, demonstrating that the proposed DOA method based on variational Bayesian inference performs well in experiments and has some improvements compared to existing methods. However, I suggest that the authors should further clarify and highlight the research contribution and innovation to better emphasize the value of the research. Specifically, in the discussion section of the article, the author can refine their language to provide readers with a clearer presentation of the results and shortcomings of the article.

2.      In the introduction section, the author can provide a more detailed background and significance of the underwater target estimation problem to better guide readers into the topic of the paper. For the references cited, the author should provide appropriate summaries. Additionally, the author should organize the language used in the article to increase readability.

Line 65 - Line 69: The array in its current state of development is was a lowcost alternative to obtain quality acoustic data from a towed array system. They demonstrate that this array can be used for observation of whales and ship tonals at ranges up to 5km. Receiving acoustic signals from target of interest with enhanced SNR and directional sensing capabilities. Marine mammal vocalizations have been captured by this prototype array and the whale species has been identified by visual observation.

3.      In the experimental section, the author presented the results of the article from three aspects and explained the advantages of the proposed algorithm. However, to improve the quality and readability of the article, the author should refine their language to express ideas more concisely, enabling readers to gain a clearer understanding of the experimental design and the results presented.

4.      The language of this article flows smoothly without too many grammatical errors, but there are some minor weaknesses in the manuscript. I recommend that the authors carefully check the paper for any grammar errors, typos, or inappropriate expressions to ensure the quality and readability of the paper.

I recommend that the authors carefully check the paper for any grammar errors, typos, or inappropriate expressions to ensure the quality and readability of the paper.

Author Response

Reviewer 3 Report (Previous Reviewer 2)

The paper is improved after revision.

Author Response

This manuscript is a resubmission of an earlier submission. The following is a list of the peer review reports and author responses from that submission.

Round 1

Reviewer 1 Report

This manuscript presents a study on the variational sparse Bayesian inference-based DOA estimation. Superiority of the method is unclear as no comparison with other methods is made. However, the authors claim in the discussion that "The proposed algorithm outperforms other algorithms in terms of estimation performance." it is not rigorous to make such a conclusion. 

Quality of manuscript needs improvement. The introduction is tedious, it is only a list of recent research without any summary. Same questions can be found in other sections. the manuscript should be reorganized to make is readable, and main contribution should be clearly written. and the conclusion is not convincing without experimental result. 

The quality of English is awful.

Reviewer 2 Report

 In this study, the author proposed a high robustness underwater target estimation method based on variational sparse Bayesian inference by studying and analyzing the sparse prior assumption characteristics of signals. So, I suggest publishing this manuscript after a minor revise. The authors did extensive work in equation on methods that the author proposed, and giving the results in both single and multiple snapshot scenarios. However, suggested to adjust in terms of language and other aspects.

As a reviewer, I have read your paper carefully and would like to offer the following review comments:

1.     Research contribution and innovation: In your paper, you clearly explain your research contribution and innovation, demonstrating that your proposed DOA method based on variational Bayesian inference performs well in experiments and has some improvements compared to existing methods. However, I suggest that you could further clarify and highlight your research contribution and innovation to better emphasize the value of your research.

2.     Language and grammar: The language and grammar used in your paper are relatively standard and accurate, but there are some minor weaknesses in the manuscript. I recommend that you carefully check the paper for any grammar errors, typos, or inappropriate expressions to ensure the quality and readability of the paper.

Line 41: You mentioned the term "MUSIC". Did you mean "Multiple Signal Classification"? You can modify it to "Multiple Signal Classification (MUSIC)" and use "MUSIC" elsewhere.

Line 47: Regarding beamforming, the author needs to make some modifications.

Line 101 and line 102: You should use "MUSIC" instead of "Multiple Signal Classification (MUSIC)".

Line 138: Please check the grammar of this sentence. "Ahmed et al. (2022) utilized the Cuckoo Search Algorithm (CSA) and swarm intelligence to optimize DOA estimation with a Uniform Linear Array (ULA) in various underwater scenarios."

Line 162: Please check the grammar of this sentence. "Mamun et al. (2020) proposed a robust prediction method using an ad hoc Bayesian Model Averaging (BMA) approach."

Line 172: Please check the tense of this sentence. "Other influential works in this field include (Yu et al., 2019), (Ma et al., 2019), and (Phoon et al., 2021)."

Line 177: Please check the term "sSBL".

Line 23 to line 187: The author listed the developments and results of many researchers in DOA estimation. The author needs to modify the tense of some sentences. For example, in line 54, the author used simple tense to introduce a result from 2019. It should be modified to past tense. Using "presented" instead of "present".

Line 211: In equation (1) and other equations, the author needs to revise the type of the variable.

Line 267: The author should add some sentences to introduce intention in using the figure.

3.     Review and citation of existing literature: You provide a comprehensive review and citation of existing literature in your paper, and critically analyze and discuss these literatures. However, I suggest that you could further improve the citation format to ensure that each citation is accurate and clear. At the same time, strengthen the analysis and discussion of the literature to highlight the differences and advantages of your proposed method compared to existing methods.

In this regard, I have identified some inappropriate areas. Please ask the author to make corrections and further check all the literature citation formats.

On the one hand, please ask the author to try to list the correct page numbers in the journal literature. On the other hand, in the literature citation at line 555, such as "Interpolat. ion" in reference 26, please confirm the correctness of the words. Otherwise, please confirm blank in line 501.

4.     Method effectiveness and reliability: In the experimental section, you demonstrated through simulation experimental results that your proposed method performs well in DOA estimation with good effectiveness and reliability. However, I suggest that you could further explore the limitations and applicability of your proposed method and explain it in detail in the discussion section.

5.     Manuscript structure and expression: The overall structure of your manuscript is clear and the expression is easy to understand, but there are some areas that could be improved. For example, you could provide a detailed introduction to the steps and theoretical basis of your proposed method in the method section. In the results section, you could add more analysis to support your conclusions. Additionally, I recommend that you follow the format requirements of the journal, including figure/table design.

Line 428: In Figure 8 (b) and Figure 8 (c), the author could use data cursors to mark the specific results, which would show the readers a clear demonstration of the simulation results.

Line 301: In Figure 3, the author could use a clearer color scheme. Additionally, please confirm the font used in the figure.

Overall, the paper provides an in-depth study of DOA estimation, and the proposed method has some innovation and practicality. However, there are still some areas that need improvement and modification. I hope that you will carefully consider my comments and suggestions and make the necessary revisions and improvements. The deficiencies indicated do not diminish the value of this review manuscript in any way. The manuscript is suitable for the journal and can be published after minor revisions.

Reviewer 3 Report

The paper proposes a high robustness underwater target estimation method based on variational sparse Bayesian inference by studying and analyzing the sparse prior assumption characteristics of signals, however, is necessary to clarify some process used to perform this analysis. Also, some factual clarifications will be helpful as highlighted in the comments below. 

- What are the conditions and characteristics of the time history shown in figure 2?  

 -Section 2 deals with narrowband signals, how is evaluated all the bandwidth signals? And what is the effect of random process? 

-The responses of Figure 5 have the same magnitude? It have been normalized?

 -Page 6, lines 263-264. The decay becomes slower as the DOF decreases. How is related with the proposal?.

-Please describe how is evaluated the Direction-of-Arrival, What is the base to define de direction? If is used any filter or is used the raw data.

-How can the corrosion degrade the mechanical response? (see suggested references)

 Please review the next papers and include it in your references.

- Fatigue damage assessment of mooring lines under the effect of wave climate change and marine corrosion, Ocean Engineering Volume 206, 15 June 2020, 107303 https://doi.org/10.1016/j.oceaneng.2020.107303

 -Fatigue of offshore structures: A review of statistical fatigue damage assessment for stochastic loadings, International Journal of Fatigue, Volume 132, March 2020, 105327  https://doi.org/10.1016/j.ijfatigue.2019.105327